

# Variation in species diversity and functional traits of sponge communities near human populations in Bocas del Toro, Panama

Cole G. Easson[1], Kenan O. Matterson[1], Christopher J. Freeman[2], Stephanie K. Archer[3] and Robert W. Thacker[4]

[1] Department of Biology, University of Alabama at Birmingham, Birmingham, AL, United States
[2] Smithsonian Marine Station, Ft. Pierce, FL, United States
[3] Applied Ecology, North Carolina State University, Raleigh, NC, United States
[4] Department of Ecology and Evolution, Stony Brook University, Stony Brook, NY, United States

Corresponding author
Cole G. Easson,
cgeasson86@gmail.com

## ABSTRACT

Recent studies have renewed interest in sponge ecology by emphasizing the functional importance of sponges in a broad array of ecosystem services. Many critically important habitats occupied by sponges face chronic stressors that might lead to alterations in their diversity, relatedness, and functional attributes. We addressed whether proximity to human activity might be a significant factor in structuring sponge community composition, as well as potential functional roles, by monitoring sponge diversity and abundance at two structurally similar sites that vary in distance to areas of high coastal development in Bocas Del Toro, Panama. We surveyed sponge communities at each site using belt transects and differences between two sites were compared using the following variables: (1) sponge species richness, Shannon diversity, and inverse Simpson's diversity; (2) phylogenetic diversity; (3) taxonomic and phylogenetic beta diversity; (4) trait diversity and dissimilarity; and (5) phylogenetic and trait patterns in community structure. We observed significantly higher sponge diversity at Punta Caracol, the site most distant from human development (∼5 km). Although phylogenetic diversity was lower at Saigon Bay, the site adjacent to a large village including many houses, businesses, and an airport, the sites did not exhibit significantly different patterns of phylogenetic relatedness in species composition. However, each site had a distinct taxonomic and phylogenetic composition (beta diversity). In addition, the sponge community at Saigon included a higher relative abundance of sponges with high microbial abundance and high chlorophyll *a* concentration, whereas the community at Punta Caracol had a more even distribution of these traits, yielding a significant difference in functional trait diversity between sites. These results suggest that lower diversity and potentially altered community function might be associated with proximity to human populations. This study highlights the importance of evaluating functional traits and phylogenetic diversity in addition to common diversity metrics when assessing potential environmental impacts on benthic communities.

## INTRODUCTION

Coral reefs are critical and dynamic habitats that provide a variety of important ecosystem services that support local economies and international industries around the world (*Moberg & Folke, 1999*; *Mumby et al., 2008*). Historically, scleractinian corals have provided the structural framework for many of these complex reef networks (*Aronson & Precht, 2001*). However, in recent decades, reefs worldwide have experienced a marked decline in the abundance of reef-building corals due to multiple stressors including marine pathogens, overfishing of herbivores, and coastal eutrophication (*Hughes, 1994*; *Lapointe, 1997*; *Hughes & Connell, 1999*; *Jackson et al., 2001*; *Harvell et al., 2007*). Caribbean reefs exemplify this trend, as many of these reef systems have undergone drastic phase shifts, resulting in the dominance of fleshy macroalgae in place of hard corals (*McCook, 1999*; *Maliao, Turingan & Lin, 2008*; *Dudgeon et al., 2010*). Along with altered community composition, structure and function (*Norström et al., 2009*), these new "stable" states provide fewer ecosystem services (*Brock & Carpenter, 2006*; *Carpenter & Brock, 2006*). Healthy reefs are typically characterized as structurally complex habitats that act as refuges for a variety of species including juvenile fish and invertebrates, effectively increasing the diversity and abundance of the associated community (*Graham et al., 2006*). In contrast, the reduced habitat complexity of macroalgal communities supports lower species diversity and productivity across numerous trophic levels (*McCook, 1999*; *Jones et al., 2004*).

In addition to increased macroalgal cover, sessile macro-invertebrates, like sponges and gorgonians, are often more abundant on degraded reefs (*Maliao, Turingan & Lin, 2008*). This increased sponge abundance may have important ecological implications for these communities (*Bell et al., 2013*), as numerous sponge species are known to perform critical functional roles on shallow reefs. For instance, sponges directly contribute to energy cycling on reefs by efficiently clearing dissolved organic carbon (*Yahel et al., 2003*; *De Goeij et al., 2013*), bacteria (*Reiswig, 1971*; *Pile, Patterson & Witman, 1997*) and pathogens from the water column, incorporating these energy sources into the benthic system (*Hadas et al., 2006*). Moreover, by hosting diverse and often abundant microbial symbionts, sponges contribute to primary productivity (photosynthesis) and nutrient cycling on the reef (*Wilkinson & Fay, 1979*; *Wilkinson, 1983*; *Wilkinson, 1992*; *Díaz & Ward, 1997*), even though microbial community composition and, subsequently, functional roles, are highly variable among species (*Easson & Thacker, 2014*; *Freeman, Easson & Baker, 2014*). Thus, as sponge abundance increases throughout the Caribbean (*Nyström, Folke & Moberg, 2000*; *Norström et al., 2009*) and populations of large species such as the giant barrel sponge, *Xestospongia muta*, increase in some regions (increased by 46% from 2000 to 2006 in the Florida Keys; *McMurray, Henkel & Pawlik, 2010*), it is likely that shifts in nutrient cycling and ecosystem function are also occurring across degraded reef systems (*Díaz & Rützler, 2001*; *Wulff, 2001*; *De Goeij et al., 2013*).

Although overall sponge richness and total biomass may be positively correlated with anthropogenic stressors (*Zea, 1994*), some sponge species may be just as susceptible to alterations in the chemical, biological and physical characteristics of the surrounding environment as corals (*Fang et al., 2014*). For example, elevated concentrations of organic pol-

lutants can influence sponge community structure by altering species diversity (*Alcolado, 2007*; *Powell et al., 2014*). Based on these data, we would predict that sponge community diversity and species composition might change across a gradient of anthropogenic stressors, but the specific response is potentially variable. Recent work has highlighted the need to move beyond simply measuring species diversity, showing that an organism's contribution to a habitat may be more important than its presence or absence (*Cadotte, 2011*; *Safi et al., 2011*; *Stuart-Smith et al., 2013*; *Stuart-Smith et al., 2015*). This principle is especially relevant in tropical ecosystems characterized by high diversity and often a high degree of functional redundancy (*Stuart-Smith et al., 2013*). Sponges represent an ideal group on coral reefs to study shifts in organismal contribution to ecosystem function, because they are prolific reef-builders, have a range of functional behaviors, and contribute a variety of crucial services to reef environments (*Díaz & Rützler, 2001*; *De Goeij et al., 2013*).

The Bocas del Toro archipelago on the Caribbean coast of Panama includes numerous islands, mangrove cays, peninsulas, fringing reefs and seagrass beds that surround shallow bays with historically high coral cover (*Collin, 2005*). The region receives high annual rainfall (3–5 m), resulting in variations in temperature, salinity, sedimentation and turbidity (*Kaufmann & Thompson, 2005*). In addition, while the Bocas del Toro region historically was home to several indigenous communities, the areas around Bocas Town and Saigon Village have recently experienced rapid large population growth (Fig. 1). This rapid population growth combined with high tourism rates, substandard public infrastructure (e.g., sewers), and deforestation has contributed to increased run-off and pollution of the near shore environment (*Aronson et al., 2004*). For example, there have been reports of "black water" outflow (sewage, road pollution and solid waste dumping) into Saigon Bay. The concentration of human activities in the Bocas del Toro region implies that some local reef communities may be negatively impacted by chronic fluctuations in water quality, while other reefs more distant from human development may be exposed to these anthropogenic stressors less frequently (*D'Croz, Del Rosario & Gondola, 2005*; *Gochfeld, Schloder & Thacker, 2007*).

The goal of our study was to build on the research of *Gochfeld, Schloder & Thacker (2007)*, who reported signs of sponge community variation that included lower diversity and higher disease prevalence near human settlement. Additionally, because measurements of alpha diversity can overlook important genetic and/or functional variability among species in the community, we also assessed potential variation in phylogenetic diversity, taxonomic and phylogenetic composition (beta diversity), and functional traits between sites, with the goal of forming testable hypotheses for how sponge community variation may translate to meaningful functional variation in communities of these increasingly dominant benthic organisms.

## METHODS

### Field sites

To assess potential sponge assemblage differences related to proximity to human development, we conducted belt transects adjacent to concentrated human settlement

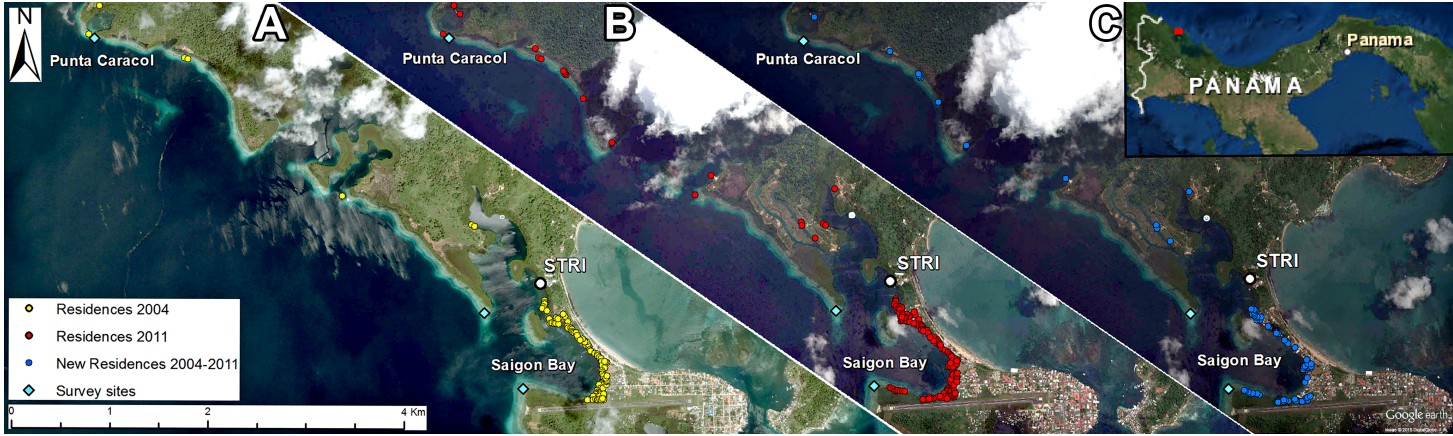

**Figure 1** **Map of Bocas del Toro region in Panama, where surveys were performed (Punta Caracol and Saigon Bay).** The white dot marked STRI represents the Smithsonian Tropical Research Institute. Each yellow dot represents an individual structure (residence) in 2004, and each red dot represents a structure in 2011. Blue dots represent new residences that were built between 2004 and 2011. Approximately 55 new structures were constructed on or adjacent to the shore along Saigon Bay in this 7-year span. Imagery is from 2004 (A) and 2011 (B and C) and is provided by DigitalGlobe® via Google Earth and ArcGIS® software imagery basemap in ArcMap™ by Esri©.

around Saigon Bay, an area where about 150 houses, an airport and several businesses are in close proximity to shore ($n = 9$, Saigon), and ∼5 km away from town where only a few houses are located ($n = 10$, Punta Caracol, Fig. 1). Surveys conducted near Saigon Bay were not in the exact same location as *Gochfeld, Schloder & Thacker (2007)*, whose site was located within the bay, likely subjecting it to naturally different environmental conditions. In the current study we conducted surveys at the mouth of Saigon Bay to better standardize reef structure between sites. Surveys were conducted at these sites in August 2012 and April 2014. Survey sites were similar in depth, exposure direction, and distance from shore. All transect data were collected on SCUBA at a depth of 5–7 m along the general axis of the reef with a minimum distance of 10 m between each transect. All specimen collection for this study was performed in accordance with a collection permit from the government of Panama, issued to Robert W. Thacker (resolution DGOMI-PICFC No. 36 issued on July 4, 2012).

## Sponge richness and diversity

Sponge diversity and abundance were characterized by counting individuals that fell within 1 m of the transect line (i.e., creating 10 m × 2 m belt transects). A total area of 200 m$^2$ and 180 m$^2$ was surveyed at Punta Caracol and Saigon, respectively. Each sponge was identified to the lowest possible taxonomic level. For sponges that were unidentifiable *in situ*, voucher specimens were collected and identified in the laboratory following spicule and fiber preparations. Using the R package vegan (*Oksanen et al., 2007*), we calculated three univariate measures of sponge diversity for each transect at each survey site: species richness ($S$), the Shannon index ($H'$), and the inverse Simpson's index ($D$). We compared each of these metrics between the two sites using a two-sample $t$-test.

## Phylogenetic reconstruction

Phylogenetic relatedness among surveyed sponge species was assessed using a phylogeny constructed from a partitioned alignment of gene sequences from GenBank coding for the small (18S) and large (28S) nuclear ribosomal subunits, which are common markers used for molecular identification of sponge species (*Redmond et al., 2013*; *Thacker et al., 2013*; Table S1). One sponge species, *Verongula reiswigi*, was not represented in GenBank, and we obtained sequence information from vouchers representing this species as part of the current study (Supplemental Information 1).

We reconstructed a phylogeny for all sponge species except *Niphates caycedoi*, for which we were unable to obtain sequence information. This species was rare, with only four individuals of this species found at one site. This species was excluded from phylogenetic analysis. For each gene, sequences were aligned using the default options of MAFFT 7.017 (*Katoh et al., 2002*) in the program Geneious (version 6.1.8, Biomatters Limited). We concatenated the two alignments, treating them as two separate partitions with independent models of sequence evolution. We implemented a relaxed-clock model in MrBayes version 3.2.1 (*Ronquist et al., 2012*), using the CIPRES computational resources (*Miller, Pfeiffer & Schwartz, 2010*), and following recommended best practices for implementing partitioned analysis (*Wiens & Morrill, 2011*; *Kainer & Lanfear, 2015*). We constrained sponges in the genus *Plakortis* as an outgroup, using the independent gamma rate relaxed clock model with a birth-death process (File S1, *Aris-Brosou & Yang, 2003*). We included three parallel runs of 10 million generations, each using four Markov chains and sampling every 100 generations. A consensus phylogeny of the three parallel runs was summarized following a burn-in of 25% (Fig. S1).

## Phylogenetic relatedness and patterns of diversity

We assessed phylogenetic diversity by calculating Faith's phylogenetic diversity (PD), using the R package picante (*Kembel et al., 2010*). Faith's PD measures the total branch length spanned by the sub-tree from each community, allowing for a comparison of total branch lengths between communities (*Kembel et al., 2010*). Additionally, Faith's PD relaxes the diversity measurement assumption that all species are "equally different" by weighting species diversity based on phylogenetic similarity (*Gotelli & Chao, 2013*). Phylogenetic diversity patterns (clustering, dispersion, or random) were assessed by measuring the mean pairwise distance (MPD) and mean nearest taxon distance (MNTD) scaled to the standard effect size among sponges within each site, accepting the default options in the models. MPD calculates the mean distance between two randomly chosen individuals in the community. Significant clustering measured by MPD implies a higher presence of species related to one another through interior nodes (away from the tips of the phylogeny) belonging to broader phylogenetic groups. MNTD calculates the mean distance separating one individual from its closest relative. MNTD describes clustering at the tips of the tree, and significant clustering by this metric indicates a higher presence of closely related species connected by nodes closer to the tips of the phylogeny. For both MPD and MNTD, we assessed differences in phylogenetic diversity patterns using two *t*-tests. We used a

two-sample $t$-test to assess differences between sites, and a one-sample $t$-test to test whether each site differed from a null hypothesis of random phylogenetic relatedness ($\mu = 0$, *Kembel et al., 2010*; *Kembel & Cahill Jr, 2011*).

## Beta-diversity analysis

We assessed taxonomic beta diversity patterns between sites by calculating Bray–Curtis dissimilarity (BCD) among all transects. We also calculated phylogenetic beta diversity among all transects, which compares MPD and MNTD between two individuals selected from different sites as opposed to individuals within the same site as previously measured (*Kembel et al., 2010*; *Kembel & Cahill Jr, 2011*). To compare taxonomic and phylogenetic dissimilarity between sites, we used the function adonis in the R package vegan (*Oksanen et al., 2007*). We used similarity percentage analysis (SIMPER) to determine the proportional contribution of each species to BCD.

## Functional trait diversity and dissimilarity

We evaluated two traits (1: microbial abundance and 2: chlorophyll *a* concentration) that are often associated with the functional roles of sponges in coral reef communities. Microbial abundance is often linked to water filtration rate. Low microbial abundance sponges (LMA) typically have higher pumping rates and thus filter more particulate organic matter (POM) from the water column. In contrast, high microbial abundance (HMA) sponges often have lower pumping rates but are able to access key inorganic nutrient sources through their symbionts. Photosymbionts represent a unique class of sponge symbionts that provide access to autotrophic nutrition and other key inorganic nutrients. Abundance of photosymbionts is often estimated by measuring chlorophyll *a* concentration within sponge tissue (e.g., *Gochfeld et al., 2012*; *Easson et al., 2014*; *Freeman, Easson & Baker, 2014*). While these two traits are often related, we assessed both traits to tease apart potential differences between HMA and low photosymbiont abundance (e.g., *Agelas conifera*, *Aiolochroia crassa*, etc.), which might occur in higher abundance at sites with lower irradiance and higher inorganic nutrients. Sponges of different classifications, with respect to these two traits, show distinct biogeochemical cycling in carbon and nitrogen cycling, which might impact the larger reef community (*Freeman, Easson & Baker, 2014*). We treated microbial abundance as a binary factor, as data for absolute microbial abundance in sponges is limited, categorizing sponges as either high microbial abundance (HMA) or low microbial abundance (LMA) based on their previously published designation (*Oksanen et al., 2007*; *Weisz et al., 2007*; *Weisz, Lindquist & Martens, 2008*; *Gloeckner et al., 2014*). We treated chlorophyll *a* concentration in sponge tissue as a continuous variable based on values in *Erwin & Thacker (2007)*. The species *Svenzea cristinae* was not analyzed in this previous survey, thus vouchers of this sample were collected ($n = 8$) and chlorophyll *a* concentration was measured using the same methodology as *Erwin & Thacker (2007)* (Supplemental Information 1). We initially compared differences in these two traits between sites using a two-sample $t$-test, assessing the proportion of HMA/LMA or High/Low chlorophyll *a* sponges (High/Low chlorophyll *a* defined in *Erwin & Thacker, 2007*) between sites. We then calculated measurements of trait diversity similarly to phylogenetic diversity
by calculating the pairwise distance among species using the values for the measured functional traits (microbial abundance and chlorophyll *a* concentration) to create a distance matrix, which allowed for comparisons of dissimilarity among co-occurring species and between sites (*Kembel & Cahill Jr, 2011*). We compared trait diversity between sites using a two-sample *t*-test to test for site differences and a one-sample *t*-test to examine whether either site differed from a null hypothesis for random trait patterns. We assessed functional trait beta diversity similarly to phylogenetic beta diversity (comparing trait distances between two individuals from different sites, *Kembel & Cahill Jr, 2011*), using the function adonis in the R package vegan (*Oksanen et al., 2007*).

### Overlap in beta diversity metrics

To investigate potential overlap among our metrics of community dissimilarity, we used Mantel tests to determine whether BCD, phylogenetic dissimilarity, and trait dissimilarity were correlated.

## RESULTS

### Field sites

Transects at Saigon and Punta Caracol contained an average of 260 and 194 individual sponges, representing $17 \pm 0.7$ and $22 \pm 1$ (mean $\pm$ SE) sponge species per 20 m$^2$, respectively. Species richness at these two sites combined for a total of 40 sponge species. Two sponges from the genus *Aplysina* were the most abundant members of these sponge communities with 681 and 587 individuals of *A. fulva* and *A. cauliformis*, respectively, pooling data from both sites. These two species accounted for approximately 28% of the total sponge individuals at each site. Other notably abundant species were *Chondrilla caribensis*, *Mycale laevis*, *Svenzea cristinae*, *Niphates erecta*, and *Verongula rigida*. Eight species (35% of unique sponge species) were unique to Punta Caracol, while 2 species (11% of unique sponge species) were unique to Saigon. These sponges were present at lower abundances within their respective community, with none of these less common species having more than 12 individuals in the entire dataset.

### Sponge richness and diversity

Species richness of individual transects ranged from 12 to 24 species. All three diversity indices were significantly different between the two sites: species richness ($S$) (mean $\pm$ SE: $17.3 \pm 1.0$ and $22.1 \pm 0.7$ for Saigon and Punta Caracol, respectively; *t*-test: $t = 3.99$, $df = 14.43$, $P = 0.001$), Shannon Index ($H'$) (mean $\pm$ SE: $2.2 \pm 0.06$ and $2.6 \pm 0.05$ for Saigon and Punta Caracol, respectively; *t*-test: $t = 4.44$, $df = 16.63$, $P < 0.001$), and inverse Simpson's Index ($D$) (mean $\pm$ SE: $7.0 \pm 0.5$ and $10.2 \pm 0.7$ for Saigon and Punta Caracol, respectively; *t*-test: $t = 3.64$, $df = 16.42$, $P = 0.002$). Saigon on average had lower species richness and community evenness compared to Punta Caracol.

### Phylogenetic relatedness and patterns of diversity

In addition to the lower species diversity at Saigon, we observed significantly lower phylogenetic diversity (Faith's PD; $3.11 \pm 0.11$ for Saigon and $3.45 \pm 0.09$ for Punta

Caracol; $t$-test, $t = 2.45$, $df = 15.39$, $P = 0.027$), indicating differences in the total branch length spanned by the sub-tree from each community. We observed no differences in MPD, between our two sites ($t$-test, $t = 0.15$, $df = 12.52$, $P = 0.873$). Although Punta Caracol displayed a pattern of random MPD (one-sample $t$-test, $t = -1.29$, $df = 9$, $P = 0.229$), Saigon showed a pattern of MPD clustering (one-sample $t$-test, $t = -3.22$, $df = 8$, $P = 0.012$). These results imply that Saigon has a slightly higher presence of more closely related species than Punta Caracol. We observed no significant differences in MNTD between our sites ($t$-test, $t = -1.40$, $df = 12.05$, $P = 0.186$), and each site displayed a random distribution of MNTD (one-sample $t$-test, $t = -0.99$, $df = 8$, $P = 0.348$ and $t = -1.94$, $df = 9$, $P = 0.084$ for Saigon and Punta Caracol, respectively). These results indicate that while phylogenetic diversity, often correlated with species richness, is lower at Saigon, these two sites do not show significantly different patterns of phylogenetic relatedness in species composition. Given the narrow geographic range of this study ($\sim 5$ km), it is possible that these patterns of phylogenetic diversity may be more indicative of the regional sponge fauna instead of differences between sites.

## Beta-diversity analysis

We observed significant differences in beta diversity for taxonomic (adonis, $F = 8.39$, $df = 1$, $R^2 = 0.33$, $P = 0.001$, Figs. 2 and 3A) and MPD phylogenetic dissimilarity (adonis, $F = 1.53$, $df = 1$, $R^2 = 0.083$, $P = 0.001$, Fig. 3B), but not for MNTD phylogenetic dissimilarity (adonis, $F = 0.69$, $df = 1$, $R^2 = 0.04$, $P = 0.476$, Fig. 3C) between sites. SIMPER analysis revealed that 5 sponge species comprised about 50% of the BCD between the two sites, including: *Svenzea cristinae* (16%), *Aplysina cauliformis* (10%), *Aplysina fulva* (9%), *Haliclona walentinae* (8%), and *Chondrilla caribensis* (7%), all of which were found in higher relative abundance at Saigon (Fig. 2). Moreover, the SIMPER results demonstrate that the lower species $D$ at Saigon was due to the increased abundance of a few sponge species.

## Functional trait diversity and dissimilarity

Analysis of trait proportions between sites revealed that Saigon had a higher proportion of HMA ($t$-test: $t = -2.63$, $df = 16.41$, $P = 0.02$) and high chlorophyll $a$ ($t$-test: $t = -9.00$, $df = 16.50$, $P < 0.001$) sponges, and the proportions of these two traits were correlated (Pearson's correlation, $r = 0.62$, $df = 17$, $P = 0.005$). Trait diversity was significantly different between our two sites ($t$-test, $t = 3.34$, $df = 10.68$, $P < 0.001$). Traits were significantly more clustered at Saigon (one-sample $t$-test, $t = -2.49$, $df = 8$, $P = 0.044$), whereas traits were more evenly distributed at Punta Caracol (one-sample $t$-test, $t = 2.73$, $df = 9$, $P = 0.006$, Fig. 3D). Species from Punta Caracol had a broad range of chlorophyll $a$ concentrations and microbial abundances, while species at Saigon were only represented by a subset of this range. Trait beta diversity between sites was significantly different (adonis, $F = 82.264$, $df = 1$, $R^2 = 0.83$, $P = 0.002$) and explained approximately 83% of the variation among transects.

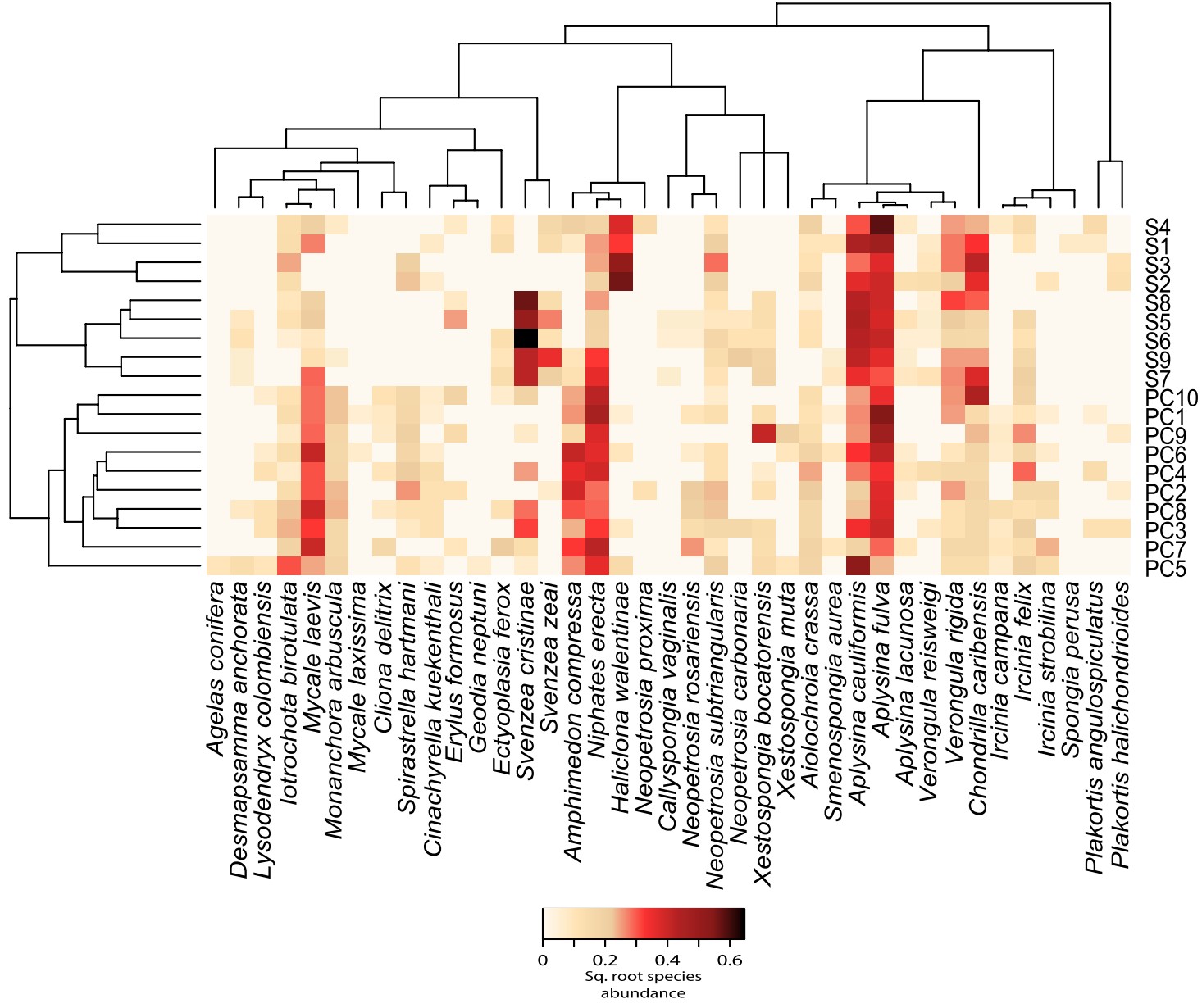

**Figure 2 Relative abundance heatmap of sponge species encountered within each transect.** These data are square-root transformed for easier visualization. A Bray–Curtis dissimilarity dendrogram on the left highlights the taxonomic dissimilarity among transects. The reconstructed phylogeny of these sponge species is displayed across the top, and species names are displayed across the bottom. S(1–9) represent transects near Saigon Bay, which are closer to a larger number of residences, while PC(1–10) represent transects at Punta Caracol.

## Overlap in beta diversity metrics

Mantel tests indicated that BCD taxonomic dissimilarity was significantly correlated with MPD phylogenetic dissimilarity (Mantel: $r = 0.48$, $P = 0.001$), MNTD phylogenetic dissimilarity (Mantel: $r = 0.51$, $P = 0.001$, and trait dissimilarity (Mantel: $r = 0.24$, $P = 0.004$). Phylogenetic MPD (Mantel: $r = 0.22$, $P = 0.010$) and MNTD (Mantel: $r = 0.27$, $P = 0.005$) dissimilarity were correlated with trait dissimilarity.

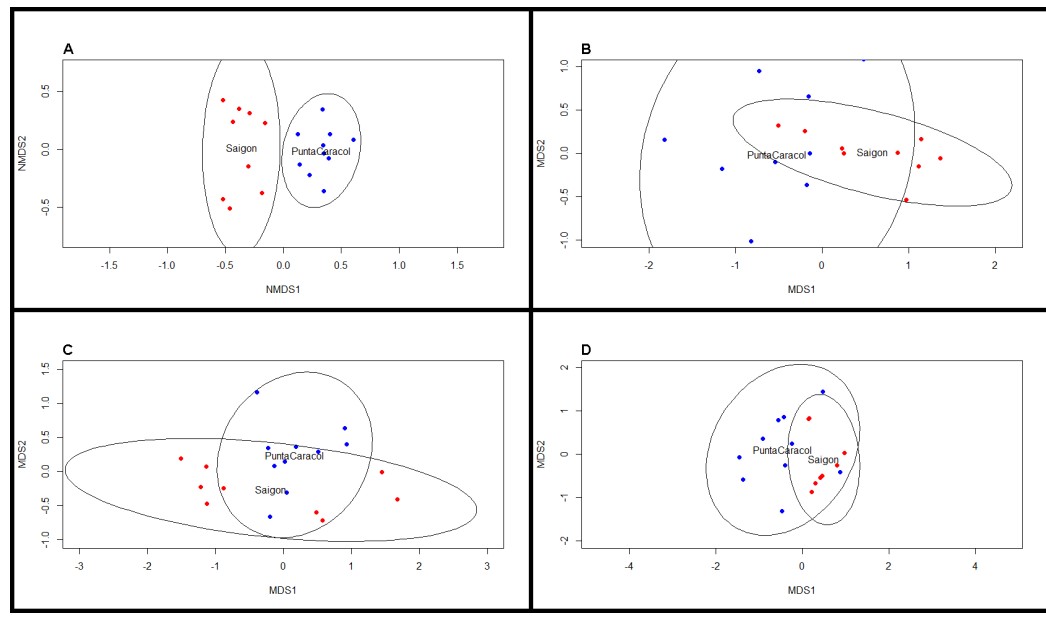

**Figure 3 NMDS scaling plots for beta diversity metrics.** Nonmetric multi-dimensional scaling plots of (A) BCD taxonomic, (B) MPD phylogenetic, (C) MNTD phylogenetic, and (D) trait dissimilarity between sites. Blue dots indicate transects at Saigon, while red dots indicate transects at Punta Caracol.

## DISCUSSION

This study identified clear differences in sponge species diversity and richness between Saigon and Punta Caracol, supporting previous findings from *Gochfeld, Schloder & Thacker (2007)*, who attributed this variation to chronic anthropogenic influence. Building on these results, we also observed significantly lower phylogenetic diversity at Saigon, and demonstrated that each site had a distinct taxonomic and phylogenetic community structure (Fig. 2). This variation in community structure resulted in contrasting trait diversity between sites, with Saigon dominated by sponges with high chlorophyll *a* concentrations and high microbial abundance.

The region of Almirante Bay in Bocas del Toro, Panama is characterized by high sponge biodiversity with over 120 species found in reef, seagrass, and mangrove habitats (*Diaz, 2005*), despite episodic heavy rainfall resulting in severe freshwater runoff, sedimentation, and low tidal flushing (*D'Croz, Del Rosario & Gondola, 2005*). Much of the development in the area has little or no sewage treatment, and many residences and businesses are built adjacent to or over the water (Fig. 1; *Collin, 2005*). Thus, many near-shore environments in this area are heavily influenced by human activity (*Aronson et al., 2004*), and abundant human debris is often observed floating and at depth in these areas (C Easson, pers. obs., 2014). Although sponges are considered to be sensitive to environmental stressors (*Gochfeld, Schloder & Thacker, 2007*), including elevated nutrient concentrations (*Easson et al., 2014*, but see *Gochfeld et al., 2012*), a factor associated with poor sewage treatment, the Bocas del Toro region maintains high sponge diversity (*Diaz, 2005*). In contrast to the high diversity and abundance of sponges in the region, coral communities have experienced

a marked decline associated with changes in water quality, concurrent with increases in human settlement in the region (*Guzman, 2003*; *Aronson et al., 2004*).

By coupling heterotrophic filter-feeding with microbial metabolic pathways afforded by their symbionts, many sponges are able to utilize a wide range of nutrient sources that likely contribute to their proliferation in a wide range of habitats, even in anthropogenically-stressed areas where corals are in decline (*Aronson et al., 2004*; *Diaz, 2005*). However, changes in sponge community diversity and composition among sites may indicate chronic environmental stress. *Gochfeld, Schloder & Thacker (2007)* found distinct shifts in composition and lower diversity associated with proximity to human development. Likewise, the current study demonstrated that sponge assemblages in close proximity to human populations were less diverse and dominated by a small number of species. When we looked beyond richness and diversity, we observed that the community at Saigon contained lower genetic diversity, selecting for more distantly related species belonging to broader phylogenetic groups. Differences in assemblage composition were driven mainly by the high abundance of *Svenzea cristinae* and *Haliclona walentinae* at Saigon; these species were nearly absent at Punta Caracol. Additionally, Saigon showed a higher abundance of *Aplysina cauliformis*, *Aplysina fulva*, and *Chondrilla caribensis*. Interestingly, these 5 species that comprise the main compositional differences between sites all have high photosymbiont abundance. In contrast to previous surveys in the area (*Gochfeld, Schloder & Thacker, 2007*), we found that the number of individuals increased at sites in close proximity to human development. This result may be skewed by the occurrence of many small individuals of *Svenzea cristinae* at Saigon (mean = 47 per transect) as compared to Punta Caracol (mean = 5 per transect). While estimates of sponge biomass or volume are necessary to definitively measure changes in sponge abundance across sites (*Wulff, 2001*), the number of individuals or species is often related to sponge biomass measurements (*Wulff, 2006*; *Wulff, 2013*).

While metrics such as diversity and richness have been widely used to estimate community health in many marine systems (e.g., *Witman, Etter & Smith, 2004*; *Schlacher et al., 2007*; *Hewitt, Thrush & Dayton, 2008*), these metrics capture only a small part of potential changes and do not consider the functional variability of species are within a habitat (*Cadotte, 2011*; *Gotelli & Chao, 2013*; *Stuart-Smith et al., 2015*). Sponges exhibit a wide range of functional behaviors, and aside from filter feeding, many of the functional roles that sponges fill are tied to the diverse and abundant microbial communities that they host. Both of the traits evaluated in the current study are related to these microbial communities (*Taylor et al., 2007a*; *Taylor et al., 2007b*). We observed significant trait diversity differences between sites, partially driven by high trait evenness at Punta Caracol. Further analysis revealed that Saigon contained sponges that were mostly HMA with high chlorophyll *a* concentrations while the community at Punta Caracol included species with a wide range of photosymbiont and overall microbial abundances. The two traits investigated in the current study are correlated because most sponges that have high photosymbiont abundance are considered to be HMA species. However, despite the long-term use of the HMA/LMA classification recent research has made several aspects of this dichotomy less clear by

demonstrating that sponge species host a continuum of microbial diversity that is specific to a species and independent of microbial abundance classification (*Giles et al., 2013*; *Easson & Thacker, 2014*). Moreover, the function of the microbial symbionts is likely more important than abundance and chlorophyll *a* concentration is a better predictor of metabolic differences among sponges than microbial abundance (*Freeman, Easson & Baker, 2014*).

Decreased trait diversity at Saigon compared to Punta Caracol could imply that ecological forces at this site are selecting for a sponge assemblage with particular functional traits. Most sponges that host photosynthetic microorganisms rely on them for nutrition to some degree (*Thacker & Freeman, 2012*), and as a result may rely less on heterotrophic feeding. We originally hypothesized that poor water quality in the region (*Collin, 2005*; *D'Croz, Del Rosario & Gondola, 2005*; *Gochfeld, Schloder & Thacker, 2007*) would lead to increased particulate matter in the water column, potentially benefitting heterotrophic sponge species and selecting against phototrophic sponges (*Weisz, Lindquist & Martens, 2008*). Instead, a higher abundance of species that host abundant symbionts, specifically, photosymbionts at Saigon may be driven by the ability of these symbionts to utilize diverse inorganic nutrient sources common in areas of anthropogenic input (*Freeman et al., 2013*; *Easson et al., 2014*; *Zhang et al., 2015*).

Chlorophyll *a* concentration and microbial abundance are somewhat limited in what they can elucidate about the sponge function, and adding more traits to the analysis would be beneficial. One trait that would provide greater resolution for sponge community function would be nitrogen transformation potential of sponge symbionts. However, data for this trait is limited to a small number of sponge species, and uncertainty surrounding the stability of these symbiont communities, as well as measurement discrepancies within a single species, prevented us from evaluating these traits in the current study (*Southwell, 2007*; *Southwell et al., 2008*; *Hoffman et al., 2009*; *Maldonado, Ribes & van Duyl, 2012*; *Fiore, Baker & Lesser, 2013*).

The current study shows that a sponge community in close proximity to human populations consisted of fewer sponge species with higher photosymbiont abundance, as well as, overall microbial abundance. In contrast, the community at a site distant from human development (Punta Caracol) included a more diverse assemblage of species, including those considered to have both high and low microbial abundance and chlorophyll *a*. Although we present no direct evidence of human impact (i.e., nutrient analysis), *Gochfeld, Schloder & Thacker (2007)* measured pollutants consistent with anthropogenic inputs in the Saigon area, and thus concluded that reefs in this region likely experience some degree of human impact. This, coupled with the fact that development in the Saigon area has continued (Fig. 1), implies that Saigon has been chronically exposed to anthropogenic inputs for at least the last 7 years. Our observations of differences in species and phylogenetic diversity, altered species composition, and functional trait diversity at otherwise similar sites suggest that proximity to human development (and potentially these inputs) may be partially shaping the community composition of these dominant benthic invertebrates. Importantly, our data also suggest that this variation may have important impacts on genetic diversity and ecosystem function. For instance,

while species with high photosymbiont abundance may increase local productivity, selection for species with abundant symbiont communities may lead to a reduction in heterotrophic feeding, instead favoring a community capable of diverse nitrogen transformations. Selection favoring HMA species over LMA species, which might rely more on heterotrophic feeding, might further alter the cycling of nutrients and organic matter within reef ecosystems by reducing water filtration rates. Thus, as shifts from coral-dominated systems to sponge-dominated systems are occurring throughout the Caribbean (*Loh & Pawlik, 2014*), it is important for us to understand how local-scale changes impact the composition of these sponge communities, as well as the functional role of species within these communities.

## ACKNOWLEDGEMENTS

We thank the staff of the Smithsonian Tropical Research Institute Bocas del Toro Research station for their assistance in the field, as well as the participants in the Bocas del Toro Sponge Course in 2012. Specifically we thank the instructors of this course, Maria-Cristina Dìaz, Eduardo Hajdu, and Gisele Lôbo-Hajdu, for helping the authors with identifying sponge species. We also thank Marc Slattery and Deborah Gochfeld for their support of this research, Julia Korn for help with data visualizations, and Taylor Roberge for manuscript feedback. All R code for statistical analyses was obtained from portions of Steven Kembel's "Biodiversity Analysis in R" workshop (http://kembellab.ca/r-workshop/biodivR/SK_Biodiversity_R.html).

### Funding

This work was supported by grants from the US National Science Foundation Division of Environmental Biology (grant numbers 0829986 and 1208310 awarded to Robert W. Thacker), the University of Alabama at Birmingham Office of Postdoctoral Education Career Enhancement Award (awarded to Cole G. Easson), Sigma Xi Grants-in-Aid (awarded to Kenan O. Matterson), and the Smithsonian MarineGEO postdoctoral fellowship (awarded to Christopher J. Freeman), the National Science Foundation award OCE-1214303 (awarded to Deborah J. Gochfeld), and the National Oceanic and Atmospheric Administration's National Institute for Undersea Science and the Technology award NA16RU1496 (awarded to Marc Slattery and Deborah J. Gochfeld). The funders had no role in study design, data collection and analysis, decision to publish, or preparation of the manuscript.

### Grant Disclosures

The following grant information was disclosed by the authors:
US National Science Foundation Division of Environmental Biology: 0829986, 1208310.
University of Alabama at Birmingham Office of Postdoctoral Education Career Enhancement Award.
Sigma Xi Grants-in-Aid.

Smithsonian MarineGEO postdoctoral fellowship.
National Science Foundation award: OCE-1214303.
National Oceanic and Atmospheric Administration's National Institute for Undersea
Science and the Technology award: NA16RU1496.

## Competing Interests

The authors declare there are no competing interests.

## Author Contributions

- Cole G. Easson and Kenan O. Matterson conceived and designed the experiments, performed the experiments, analyzed the data, contributed reagents/materials/analysis tools, wrote the paper, prepared figures and/or tables, reviewed drafts of the paper.
- Christopher J. Freeman performed the experiments, wrote the paper, reviewed drafts of the paper.
- Stephanie K. Archer conceived and designed the experiments, performed the experiments, reviewed drafts of the paper.
- Robert W. Thacker analyzed the data, contributed reagents/materials/analysis tools, reviewed drafts of the paper.

## Field Study Permissions

The following information was supplied relating to field study approvals (i.e., approving body and any reference numbers):

The Republic of Panama

Resolution DGOMI-PICFC No. 36 issued July 4th, 2012.

This permit was issued to Robert W. Thacker, and allows for the collection of marine organisms for scientific purposes at the Smithsonian Tropical Research Institute at Bocas del Toro, Panama.

## DNA Deposition

The following information was supplied regarding the deposition of DNA sequences:

GenBank accession numbers can be found in Table S1.

## Supplemental Information

Supplemental information for this article can be found online at http://dx.doi.org/10.7717/peerj.1385#supplemental-information.

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
