# Peer review of "Variation in species diversity and functional traits of sponge communities near human populations in Bocas del Toro, Panama"

_PeerJ, doi:10.7717/peerj.1385_

## Round 0.1 · original submission · Major Revisions

· Academic Editor

Major Revisions

I now have 3 reviews back on your manuscript, and all have substantial criticisms of the manuscript. The first recommends rejection based on the technique employed, and each of the others provide extensive feedback on the writing, analyses and presentation of the results. I am unconvinced that the study is fundamentally flawed and needs to be redone simply because of the transect method employed, but given that feedback it seems something that the authors will want to address. I am confident that you will have an opinion on that criticism that I look forward to seeing it. The criticisms provided by the other referees is extensive, in particular with reference to the details of the experimental design and the previous work by Gochfeld et al. that should be clearly addressed by the authors in the revision. Having said that, my reading of the reviews by the second and third referees leads me to believe that the manuscript may become acceptable with suitable revision, and so I would like to send a revised manuscript back to them for additional feedback when you are ready.

Reviewer 1 ·

Basic reporting

Belt transect should be the appropriate method to use, instead of transect line. Transect line underestimates smaller size sponges and species. This distorts the real sponge community structure or the pattern of species dominance, and species richness component is underestimated. Transect is better used for assessing cover but not community composition and structure indices based on counting individuals. It skews towards greater size species.
Results using belt transect very probably would have provided very different results (smaller or medium sized species would have been better represented, and species diversity or species richness would be more representative. Very probably some of present species were omitted.
For that reason, the validity of results dealing with Taxonomic and Phylogenetic Dissimilarity, chlorophyll a concentrations and microbial abundances is doubtful.

Experimental design

Appropriate experimental design, but wrong sampling methodñ

Validity of the findings

Non valid because using tansect line sampling, which skews toward bigger sizes. V ery probably results could be skewed too.

Additional comments

I recommend resampling the study sites using belt transect to make research valid.
See more comments in the text.

Reviewer 2 ·

Basic reporting

Overall, the writing and presentation are well done, with the following notations/suggestions:

Ln 49 discusses ‘changes in sponge communities’, however the present study does not look at change over time but differences between two sites. I would reword for clarity (though see comment below about assigning differences to variation in human population).

Ln 237: The authors present species richness in the order Saigon then Punta Caracol, however the following results are presented in the opposite order (Punta Caracol then Saigon). Keep order consistent for easier interpretation.

The methods note that 20 m2 transects were performed, and all results are presented as number per transect. Why not present these as number per 20 m2?

While the visualization of relative abundance in Fig 2 is useful to look at, I would appreciate a table of actual values, either as raw value per transect or mean and standard error.

I would also repeat the Transect number on the left side of Fig 2.

Experimental design

The authors attempt to address the hypothesis that sponge communities vary on a gradient of anthropogenic stressors. The hypothesis is then tested by comparing two sites of varying proximity to human inhabitants; however there is no data to support that the two sites indeed vary in any anthropogenic stressor. The only evidence provided is a previous study (Gochfeld et al 2007), however in that study there was no difference in the measured variables (nutrient concentrations, fecal coliform, and PAHs) between the two study locations used in the present study. Thus there is no evidence for any gradient, and therefore any observed differences cannot be assigned to differences in anthropogenic stresses. The paper should be restructured in response.

The following information should be included regarding field surveys:
• When were surveys conducted?
• Were all surveys conducted in the same time frame?
• How was variation in surveyor accounted for or standardized?
• How far apart were transects at each site?
• Is it possible to provide GPS coordinates?
• Are transect numbers in Fig 2 geographically closer together?
• Methods indicate that 10 transects were done at Punta Caracol and 9 at Saigon. However, Line 237 in the results states there was 1 survey at Saigon and 3 surveys at Punta Caracol. A clear distinction needs to be made regarding transects and surveys. Were the 10 surveys at Punta Caracol done at 3 different times or locations? Or were the 10 transects repeated 3 times at Punta Caracol, but only once at Saigon? Please clarify.
• Were assumptions of statistical tests accounted for and any data transformations done?


Trait descriptions:
• A supplemental table of HMA/LMA attribute and chl a data should be provided.
• The authors state that they consider chl a to be a continuous variable but then group species into high and low chl a concentration. What cut off was used to make this distinction?

Validity of the findings

The comments in the Experimental Design portion make it difficult to assess the robustness of the dataset as well as what conclusions can be drawn based on the observed patterns.

Species diversity (which is correlated with phylogenetic diversity) differed between the two locations. However, this appears to be due to a few species at Saigon site (Ln 267; 321). These differences could be due to recruitment events of individual species or even variation in predator abundances. Interestingly, the two species that comprised the difference (Svenzea cristanae and Haliclona walentinae; Ln 321) were not observed at either site in Gochfeld et al 2007. It appears that Svenzea cristanae (which I can only assume is the Svenzea sp presented in Figure 2, as there is no Svenzea cristanae in the figure) was found in high abundance at 5 of the Saigon sites. Differences between the present study and Gochfeld et al 2007 may also be in identification, but should be addressed by the authors.

In addition, two species of Aplysina (A. fulva and A. cauliformis) which made up 19% of the dissimilarity were not observed at Saigon in Gochfeld et al 2007, yet appear in high abundances in the present study. Both species are relatively slow growing species. Since the differences between sites appear due to a few species, discussion of recruitment and other abiotic and biotic interactions beyond anthropogenic stress tolerance is warranted.

Given that trait diversity is linked to species diversity, then the additional factors suggested here should be considered as selective factors as well. Interestingly, the Saigon site with the suggested higher anthropogenic stress had higher proportion of chl a species. One might predict that increased nutrients and runoff would decrease light availability and thus photosynthetic ability and you might actually see fewer species in this habitat.

Finally, as the authors note the present study does not include any details on size of individuals which is extremely important to consider when discussing functional roles of specific species and overall ecosystem functioning. This further limits the conclusions that can be drawn.

Additional comments

The present study represents a great effort in relating phylogenetic diversity with abundance in a community and attempts to link ecological function as well. While there appears to be some interesting patterns and differences between the two communities, based on the presented data it is impossible to discern why these patterns exist. The paper should be restructured to focus on the diversity-trait connections, with some limited discussion on why these variations in the two communities exist.

Reviewer 3 ·

Basic reporting

The manuscript is mostly written well, but there are a lot of wordy and confusing sentences. There is also a high amount of repetition of words within a sentence, which could be avoided if they were written more succinctly. There is a very short, but exceptionally useful book called “Elements of Style” by Strunk and White, which is an excellent guide to scientific writing.

There is no mention of Figure 4 in the manuscript.

I would suggest that the author includes all R scripts in the supplementary materials so the methods are repeatable.

The Chlorophyll a concentrations calculated for all the sponges need to be included in the supplementary materials.

The title could be better written to reflect the inclusion of phylogenetic relatedness and beta diversity etc. Also there are only 2 sites, so it’s not really a gradient, especially and the chosen sites may not even represent the extremes at both ends of the anthroprogenic disturbance scale.

The author needs to move all collection permits to the methods section.

Experimental design

See General comments

Validity of the findings

See General comments

Additional comments

An interesting and important piece of work with relatively novel and informative analyses. Needs some work in reordering and expanding on points, but will be a good paper in the end.

Abstract:
Line 35: Please add in how many sites were surveyed, this will help the reader get a better idea of your study design from the abstract.

Line 41: I would suggest describing where Saigon Bay sits in the range of proximity to human development, much like you did for Punta Caracol.

Lines 44-49: Without having read the rest of the manuscript yet these sentences are a little confusing and wordy I’d suggest reworking them to clarify these points.


Introduction:

Line 60: Are there not more recent citations for this point?

Line 61: Please add a citation to support this.

Line 72: I would suggest that the writer reduce the wordiness in this sentence. Perhaps instead write something like “In contrast, the reduced habitat complexity of macroalgal communities supports lower species diversity and productivity across numerous trophic levels (McCook 1999, Jones et al. 2004)”

Line 92: there are 2 closing brackets after the Zea 1994 reference.

Line 104: How recent was this recent population growth?

Line 110: Can you explain in a little more detail what the environmental differences are between these sites in the methods, like phosphates, nitrates etc.

Line 112: This sentence is confusing after “…are in close proximity…”. I would split this sentence or maybe leave this detailed description for the methods and describe broadly what Saigon Bay is like i.e. low or high anthropogenic impact.

Line 115: Please explain why you chose to look at these measurements e.g. relatedness/function etc to determine whether the anthropogenic pressures were impacting the sponge assemblages.

Line 118: consider revising to …”we compared sponge assemblage richness and diversity to patterns of phylogenetic relatedness”

Line 123. Move this last sentence to the methods. Perhaps at the end of the introduction write one sentence bringing it all back to the big picture, i.e. why this is important work.

Line 124: missing a period at the end of the sentence.

Methods:

The methods could do with some re-writing and re-structuring to make it clearer. It’s a little confusing at times with some sentences don’t appear to flow from the one before and therefore don’t make much sense.

Also I’m not sure why these methods were used to answer these questions. Why would you reduce multivariate data down to univariate analyses? I’m not saying you shouldn’t have used the methods you did, but perhaps explain why you chose to use them.

Currently your 3 main questions at the end of the introduction are:
1) we compared the richness and diversity of the sponge community between sites, and tested whether these differences in diversity could be attributed to patterns of phylogenetic relatedness.
2) we compared beta-diversity between sites by considering species composition, relative abundance, and phylogenetic relatedness.
3) we assessed whether the relative abundance of sponges possessing distinct functional traits differed between sites.

And the subheadings in the methods are:

Field surveys
Chlorophyll a analysis
DNA extraction
Reconstructing Phylogenetic Relationships
Species diversity
Phylogenetic patterns of diversity
Taxonomic and Phylogenetic Dissimilarity
Trait diversity and dissimilarity

I would re-organize the sections (methods, results and discussion) so that they follow the same order and similar wording to the 3 main questions at the end of your introduction so the reader is lead through in a consistent fashion.

Looking at your 3 main steps at the end of the introduction I would suggest something like: (and the order can change if you change the order of your questions)

1. Field sites (here you write about the locations of the sites and the environmental conditions)
2. Sponge richness and diversity (detail how you collected the data and analyzed it)
3. DNA sequences and extraction (which samples did you get from GenBank and which did you personally need to sequence and how)
4. Phylogenetic relatedness and patterns of diversity (combine “reconstructing phylogenetic relationships” and “phylogenetic patterns of diversity” together)
5. Beta-diversity analysis (here put the “taxonomic and phylogenetic dissimilarity” section, basically how you did this step by step by considering species composition, relative abundance, and phylogenetic relatedness) –maybe move this question to the 3rd one so it comes after functional traits and you can then move line:219 to the end of this section as well.
6. Functional trait diversity and similarity (include sponge related trait measurements (chl-a and microbes), how you collected the data, why you picked them and how you analyzed them i.e. combine and add to “Chlorophyll a analysis” and “Trait diversity and dissimilarity”.

Line 127: Did one person do all the surveys?

Line 129: need a space between creating and 10 and “each individual” is a redundant phrase, consider removing one of the words. It might also be nice here to say how many m2 of reef in total were surveyed per site.

Line 132: Clarify that you are talking about chlorophyll a levels in sponges, as often this is a water column measurement.

Line 133: Can you please explain here or the introduction how measuring chlorophyll a (and microbes) can be linked to functional traits in sponges and why it’s important to look at specifically in this study. Also describe how you obtained both the chl-a AND microbial data. The first mention of microbes after the introduction is when this information is included in the trait diversity analysis and it’s not clear where you obtained this data and what format it’s in.

Line 142: Did you use standards when making these measurements?

Line 148: Can you please briefly add why you chose these markers. 18s is not normally the best marker for sponge species level differentiation.

Line 155: What reagents (including concentrations and volumes) did you use for your PCR and under what conditions were they run in the thermocycler?

Line 162: Did you trim the sequences to the same length?

Line 164: I’m a little confused here. The DNA extractions in the previous paragraph refer only to one species, but now you are talking about several species. Please clarify in the previous paragraph how many species you include in the phylogenetic analyses and where you obtained their sequence information. Did you just pull all other sequences apart from Verongula reiswigi from GenBank?

Line 169: How did you decide to use this model?

Line 184: What’s the difference between the MPD and MNTD measurements?

Line 206: Do you mean similarity?

Line 211: missing “on” between “based” and “their”

Line 214: I’m a little confused as to why you now refer to this single species, especially if the method to obtain chl-a is the same as you previously described.

Line 216: missing “test” or “examine” or something similar between “to” and “whether”

Line 217: “We initially compared” makes it sound like you there should be a follow up like “and then we…”

Line 219: please provide more information as to how you assessed beta-diversity as it can refer to a number of indices inferring compositional heterogeneity. Also this sentence seems a little out of place.

Line 222: Is this sentence meant to be a paragraph on its own, again it seems out of place.

Results:

Line 237: this should probably just go in to the field survey results.

Line 238: include in full the 3 diversity indices you measured next to the symbols

Line 243: I can see what you’re saying but this sentence is written in a confusing way. Using “but also” sounds weird and suggests that the finding of a few species dominating in an ecosystem is unusual when in fact this distribution pattern of species abundance is common in ecology.
You’ve also already pointed out above that Saigon has fewer species so here just talk about the difference in evenness between the sites.

Line 259: please add the distance between the sites here. This sentence is also more of a discussion point and shouldn’t be in the results section.

Line 269: wouldn’t a lower species dominance (D) be the result of fewer dominating species as opposed to an increased abundance of them?

Line 279: Isn’t this just repeating what you just wrote above?

Discussion:

Line 291: Didn’t Gochfeld et al 2007 find the same result? This paper supports Gochfeld’s findings it doesn’t just use the documented gradient of anthropogenic disturbance from that paper. This work appears to be a continuation of the Gochfeld et al 2007 paper, and there seems to be a little bit of overlap between these studies. I’d make it clearer throughout that this work is a continuation of this previous study (but still independent work) and that the current research is looking more at phylogenetic and trait differences and why it’s important to do so.

Line 293: Punta Caracol “is” further away (not "was", unless its circumstances have changed)

Line 299: take “the” from between “in” and “reef”

Line 311: contribute not contributes because you a talking about multiple nutrients not just the one. Also need a comma after diversity and pathways. Also need at least one citation here.

Line 313: Try removing a couple of uses of the word “community” in this paragraph, if there is diversity it’s going to be based on a community (or more accurately an assemblage) of sponges.

Line 316: Again it’s quite normal to find systems which are dominated by a few species, so why is this interesting? It might be because of those particular species that dominate.

Line 320 and 322: Are these dominant species normally found in this region and/or introduced species which are the taking advantage of the disturbed conditions? Are they species normally found in low water quality conditions?

Line 323: this could be a seasonal pattern, did you survey at the same time of year as Gochfeld? Or due to a recent disturbance (i.e. a storm) which fragmented the sponges.

Line 331: I would stress this more in the introduction, make it one of the main reasons you are doing this study. Maybe move some of this paragraph up to the introduction.

Line 341 and 343: these sentences sound like they should be in the introduction.

Line 348: This is a confusingly written sentence “indicating its increased species diversity TO sponges…”

Line 353: I think instead of “Additionally…” you should write “However, despite the long-term use of the HMA/LMA classification recent research…” (or something similar, because the use of additionally sounds like it’s another point against using both microbes and chl-a together whereas you are now describing why it’s good to. Also why didn’t you use a continuum of microbial abundance in this study instead of the HMA/LMA system? If it wasn’t possible to use a continuum say so in the methods.

Line 358: It would be nice to know why exactly you decided to include both in the methods.

Line 361: I’m not sure exactly what you mean by “with a SUBSET of the overall trait diversity” please clarify.

Line 391: This work supports the notion, it’s not the first time this relationship has been found.

Line 395: I wouldn’t write “although”, it’s precisely because the coral reefs have turned to sponge reefs in the Caribbean that makes it’s particularly important to study sponge communities here. I would also add a point as to why looking at phylogenetic relatedness is also particularity informative.

It would be nice if Figure 2 was considered more in the discussion

---

## Round 0.2 · Minor Revisions

· Academic Editor

Minor Revisions

Both referees agree that your revisions have greatly improved the manuscript, and that the manuscript will be suitable for publication following minor revision. The one remaining point of contention is the treatment of the sequences in the phylogenetic analysis. The referee contends that the sequences must be trimmed to equal length to avoid biasing the results, but you contend that this does not matter.

I have always trimmed sequences to equivalent length myself, but had to admit that I had never done the experiment to determine whether or not it altered the results, and I did not want to return a decision based on my belief. So I asked my post-doc to try it on a couple of data sets we had handy, and he found it could alter the results, depending on the data set - and the results could be profound when the number of informative sites in the columns that have no missing data was lower than the columns that did have missing data. So, I followed up with two of my colleagues who are experts in phylogenetics and asked their opinion. One replied that it is widely accepted to trim sequences, although they could not think of any really strong papers that justify the practice. They went on to recommend I read Lemmon et al. 2009 Syst. Biol. 58:130-145. I did, and in that paper, they report cases where you can get serious biases (particularly for branch length estimation) when the missing data is distributed non-randomly with respect to rate. The other colleague I spoke to explained that if we have an alignment that only varies at one nucleotide position (and say that position is fixed in one species near the edge of the alignment), then all samples without that position will fail to cluster it with other samples, and they will be pulled more basal towards a star or unresolved polytomy in the phylogeny. Their opinion was that it would quite easy for several homoplasious sites in sequrences of unequal length to overwhelm the signal.

Based on these discussions, I have to go with my initial reaction and conclude that the burden is on the authors to convince the referee and futrure readers that missing data is not a problem. I would like to see a detailed explanation of why the particular method they are using is not sensitive to missing data, or an additional comparison, either by permutations of the data, or by comparing the results of trimmed and untrimmed sequence analyses, to document that this does in fact make no difference to the results.

Reviewer 2 ·

Basic reporting

No Comments

Experimental design

No Comments

Validity of the findings

No Comments

Additional comments

The revisions and comments provided adequately address previous concerns and the current MS meets all criteria for publication.

Reviewer 3 ·

Basic reporting

No comments

Experimental design

No Comments

Validity of the findings

My only concern is with the response to one of my comments:
Line 162: Did you trim the sequences to the same length?
Authors response: The sequences were not trimmed to the same length. Such steps are not required for phylogenetic analysis.
My response:
It may not be required to have all sequences the same length for phylogenetic analyses, but if they are not this can have a profound influence the result. If a large portion of your informative sites are not represented across all taxa, then it can severely bias the results. Some methods are more sensitive to this than others, and for some data sets it will not matter. However this needs to be demonstrated to the readers.

Additional comments

A marked improvement has been made to this manuscript. It now reads considerably better and the findings are more apparent. Overall a nice paper, which adds useful information to the body of Porifera research.

I have only a few small comments:
Introduction:
Line 57: Suggestion: change ‘stony’ to ‘scleractinian’
Line 75: Needs a general citation here (e.g. Bell et al, 2013 and/or Bell, 2008 and/or Wulff, 2006)
Methods:
Line 138: data is plural, so it should be “all data were”
Line 154: This and the next paragraph (from line 161) sound very similar and repetitive at the beginning, do they need to be combined or made more distinct from each other?
Line 186: I don’t know this software, but wouldn’t significant clustering, which implies higher relatedness among samples be indicative of species belonging to a narrow range of phylogenetic groups as opposed to broader? Please clarify if it’s not a typo.
Line 209: missed “to” between “but are able” and “access key”

Discussion:
Line 391: “higher photosymbiont”, should be maybe “higher photosymbionts”? Or “with a high proportion of photosymbionts or similar.

Line 393: suggestion: avoid repetition of the same word in a sentence. replace the repeat of “species” for “those".

I think one of the interesting points in this paper is that even though Saigon Bay is considered the high population/disturbance site, globally this is probably considered low, but you are still seeing impacts on the sponge assemblages.

---

## Round 0.3 · accepted · Accept

· Academic Editor

Accept

Thank you for your constructive responses to the referee comments, and for passing along the Weins & Morrill paper, which I had missed. Having read both that and your response, I am satisfied that you have considered the issue carefully, and am happy to accept the manuscript. However, I also have to point out that it is not just the referee and I who had this impression - the two colleagues I consulted as experts in the field (one a microbial ecologist and the other who works on vertebrates and is a well-known theoretician) also had the same initial impression as I did. Given that 4 of your potential audience shared the same misgivings (whether justified or not), I expect that your typical readership is very likely to have a similar response. Thus, I believe it would be to your benefit to include at least some of that text to clearly justify your approach and your reasoning in this matter included with the Bayesian tree in the supplementary materials to prevent future readers from asking the same questions. However, this is a personal decision which I leave up to you as the authors, and whether you decide to include any of the rebuttal text in your supplementary materials or not, I am satisfied with your revision and am willing to move the manuscript forward.